# Efficacy of Continuous Suctioning in Adenoidectomy Haemostasis—Clinical Study

**DOI:** 10.3390/medicina59091534

**Published:** 2023-08-24

**Authors:** Veronica Epure, Razvan Hainarosie, Dan Cristian Gheorghe

**Affiliations:** 1ENT Department, Carol Davila University of Medicine and Pharmacy, 020021 Bucharest, Romania; 2ENT Department, “Marie Curie” Children Hospital, Bd. C. Brâncoveanu 20, 041451 Bucharest, Romania; 3I.F.A.C.F.-ORL Prof Dr. D. Hociota, M. Cioranu 21, 061344 Bucharest, Romania

**Keywords:** operative time, haemostasis time, adenoidectomy, continuous suctioning, primary postadenoidectomy hemorrhage

## Abstract

*Introduction*: Adenoidectomy is often the first major surgical challenge for the child’s haemostatic system, and controlling intraoperative bleeding can be a challenge for the surgeon. Different methods have been used intraoperatively by surgeons in order to enhance haemostasis. The cold air effect (continuous suctioning) has been used by some surgeons during adenoidectomy; however, no documentation of its haemostatic effect has been made. *Objectives*: Our prospective randomised controlled study enrolled a sample of 140 children undergoing adenoidectomy, and we studied the effect of continuous suctioning on the duration of haemostasis in paediatric adenoidectomy. *Materials and Methods*: We evaluated the effect of using continuous suctioning during haemostasis at the end of adenoidectomy procedures, comparing variables such as total surgery time, total haemostasis time, and intraoperative blood loss, between two groups: 70 adenoidectomy procedures where no continuous suctioning was used to enhance haemostasis versus the other 70 patients where continuous suctioning was the haemostatic method employed. RESULTS: After statistical analysis of the recorded data, we found that the total duration of adenoidectomy, the duration of haemostasis in adenoidectomy, and the intraoperative blood loss were significantly lower in patients in whom cold air was used for haemostasis. Intraoperative haemostasis failure (and consequent use of electrocautery for haemostasis) was more frequent in patients in whom no suctioning was used; as for the rates of postoperative primary bleeding after adenoidectomy, they were similar in both groups of patients, regardless of the technique used for haemostasis. *Conclusions*: The use of continuous suctioning during adenoidectomy haemostasis significantly shortens total surgical and haemostasis time, reduces intraoperative blood loss, and reduces the incidence of haemostasis failure (with the consequent need for bipolar electrocautery haemostasis).

## 1. Introduction

Adenoidectomy is one of the most commonly performed surgical procedures in paediatric ENT [1,2]. In children, these procedures are sometimes the first major surgical challenge to their haemostatic system. Adenoidectomy is indicated for obstructive sleep apnea or recurrent infections of the adenoid mass and neighbouring organs [1]; adenotonsillectomy’s incidence has dramatically increased over the last 20 years due to its utility in preventing obstructive sleep apnea and its negative side-effects (cognitive and behavioural disorders, learning difficulties, nocturnal enuresis, hypertension) [2].

Although generally accepted as a safe procedure by most surgeons, postoperative bleeding remains the most significant and scarring complication of adenoidectomy. Postadenotonsillectomy (PAT) bleeding, although rare (cited incidence of 0.5–4.2%) [1,2], requires immediate treatment and can be life- threatening to the patient, whereas intraoperative bleeding remains a challenge to the surgeon, requiring immediate action to control it.

Decreasing postoperative pain, reducing the operative and haemostasis duration, and minimising intraoperative and postoperative haemorrhage risk have brought attention to new surgical techniques and instrumentation. Different techniques are nowadays used for paediatric adenoidectomy: cold surgery—the conventional curettage (so-called blind curettage) with Beckmann curette; microdebrider (shaver) or endoscopic—assisted adenoidectomy; hot methods—electrocautery adenoidectomy [3,4], adenoidectomy with radiofrequency (coblation) [3,4], each method presenting its own advantages and possible side-effects.

Most authors agree that operative time and blood loss are significantly reduced in the case of electrocautery used in adenoidectomy surgery [5,6,7,8]. Whereas some authors report no complications after electrocautery adenoidectomy [9,10], others report a higher risk of secondary (delayed) haemorrhage and a greater incidence of neck pain (referred pain at the back of the child’s head) after bipolar electrocautery use (incidence up to 12%) [2,4,5,11], prolonged nasal obstruction, and velopharyngeal insufficiency symptoms.

Most authors agree that electrocautery use during adenoidectomy or adenotonsillectomy, regardless of duration of use, produces notable side-effects such as delayed epithelisation of the surgical bed [5,6,7,9,12]; all electrosurgical lesions demonstrate some adjacent areas of desiccation, coagulation, and carbonisation, regardless of the power, duration, or waveform used (radiofrequency included) [11,12,13].

Knowing the worrisome side-effects of electrocautery use in performing haemostasis for adenoidectomy, alternative methods to reduce blood loss and haemostasis time in adenoidectomy were investigated. Surgeons have started to use different biomaterials or topical agents at the end of surgical procedures in order to speed up haemostasis; however, there are always issues related to materials’ availability or costs and the personal preferences of the surgeon. Haemostasis during adenoidectomy can be spontaneous, aided by gauze packings in the nasopharynx (simple or with vasoconstrictor agents as pseudoefedrine, xylometasoline, epiephrine dilution [10], tranexamic acid [14]), irrigation with hydrogen peroxide [15] or saline solutions at different temperatures [16,17], applying fibrin-based biomaterials or oxidised cellulose [18,19]. Applications of bipolar electrocautery or posterior nasal packing [19] are used as final solutions when all previous methods for haemostasis fail. Continuous suctioning has been used by some surgeons during adenoidectomy haemostasis; however, no documentation of its haemostatic effect has been made.

While various haemostatic techniques used during adenoidectomy have been studied [8,12,13,14,15,16,17,18,19], there is limited evidence of the efficacy of continuous suctioning during adenoidectomy haemostasis [20,21,22,23,24,25,26]. This study refers to intraoperative postadenoidectomy bleeding and ways to control it effectively. The aim of this study was to evaluate if continuous suctioning during adenoidectomy (creating a local “cold air effect”), a costless and easily accessible method, is efficient for shortening haemostasis and operative time and reducing intraoperative blood loss. The effect of suctioning used intraoperatively on the incidence of early postadenoidectomy haemorrhage was also studied in our enrolled children.

## 2. Methods

We conducted a prospective randomised controlled study in the ENT Department of a tertiary Children’s Hospital between November 2022 and January 2023, involving children undergoing adenoidectomy. The aim of our study was to assess the cause—effect relationship between an intervention (use of continuous suctioning during adenoidectomy) and outcome (duration of haemostasis, total operative time, intraoperative bleeding, and incidence of early postoperative bleeding).

Our Hospital’s Ethics Committee has approved this clinical study (53792/12 December 2022). After explaining the procedures and objectives of the study, parents’ written consent was obtained in each case for preoperative blood tests, surgery, general anaesthesia, and inclusion in this study.

In our study, patient records were anonymised, data were collected in an Excel database, and it was analysed. The investigators for this study introduced in the OR protocols for adenoidectomies the exact measurements of the operative time, haemostasis time, and number of gauze packings used intraoperatively for each adenoidectomy procedure. The use of bipolar electrocautery to complete haemostasis was recorded for each patient in their surgical protocol.

Inclusion and exclusion criteria. 140 children were included in this study, divided into two groups. Children aged 0 to 18 years were enrolled if they met the following inclusion criteria: 1—child admitted to the Paediatric ENT Department of our hospital; 2—indication of adenoidectomy under general anaesthesia (due to obstructive sleep apnea or severe recurrent upper respiratory tract infections; recurrent acute otitis media or chronic serous otitis media); 3—laboratory tests (blood markers) performed preoperatively by our hospital’s laboratory with no major coagulation abnormalities reported; 4—simple adenoidectomy procedures performed between November 2022 and January 2023 by the same 3 ENT experienced consultants; 5—adenoidectomies performed only on 2 days per week (Wednesdays and Fridays) during the trial period.

Adenoidectomies performed by any surgeon in the department on other weekdays were excluded. Adenoidectomies with myringotomy tube insertion or adenotonsillectomies were excluded from our study. Adenoidectomies performed by other surgeons, even if during the same 6-month period in the same department, were excluded.

The two analysed groups were defined by the presence or absence of a specific intervention (use of continuous suctioning). Group A enrolled children for whom continuous suctioning was used at the end of the adenoidectomy procedure. Group B consisted of children in whom no continuous suctioning was used during an adenoidectomy. Children were randomly recruited by the leading investigator from children undergoing adenoidectomy in our department over a 6-month period and assigned randomly to one of the two groups.

Patients were randomly assigned to groups A or B preoperatively. The leading investigator chose intervention days such that all enrolled surgeons used continuous suction on these specific days, while they used other haemostatic methods during the rest of the study (cotton packings, electrocautery). As a result, every surgeon out of the three participating in our study performed adenoidectomy with or without the application of continuous suctioning during this study period, but on different days. This was decided in order to eliminate bias and ensure the randomisation of the participants.

All adenoidectomies were performed under general anaesthesia with orotracheal intubation by three different surgeons, consultants with over 15 years of similar surgical experience, using the same technique and identical equipment, as well as the same anaesthesia procedure. General anaesthesia was performed with sevoflurane induction, and maintenance is carried out accordingly with intravenous propofol along with sevoflurane as needed. After initial endoscopic evaluation of the child’s nasopharynx (drug-induced sleep endoscopy), classic cold instrument adenoidectomy was performed in all cases (Beckmann’s curette adenoidectomy technique) under direct vision (hyperextension of the child’s neck, unilateral Nelaton’s probe used to retract the soft palate anteriorly and cranially for enhanced direct visualisation of the nasopharynx). Digital palpation of the adenoid mass and nasopharynx was routinely performed prior to and after the procedure in order to ensure complete excision of the adenoid mass. Endoscopic control is always used to assist adenoidectomy when local anatomy makes the inspection of the child’s upper rhinopharynx (near the choanae) impossible. After complete resection of the adenoid mass, haemostasis follows these steps: a cotton packing is introduced in the nasopharynx twice (each left in place for a maximum of 1 min), then another packing is soaked in vasoconstrictor medication (epinephrine dilution 1/10), then direct irrigation with saline solution at room temperature (approximately 10 mL). In patients from group A, continuous suctioning via a metallic probe for 3–5 min was applied to the nasopharynx, set at 150 mbar/hPa. No action was taken in group B patients—cotton packings (without other haemostatic solutions) were used as needed until local bleeding stopped or the surgeon decided to use electrocautery for haemostasis.

No other haemostatic materials such as hydrogen peroxide, xylometasoline, topical tranexamic acid, or fibrin-based haemostatic powder were used either in groups A or B to enhance haemostasis. In cases of failure of haemostasis (over 10 min of haemostasis duration, failure of the saline irrigation solutions to clear or localise persistent bleeding from the nasopharynx), the consequent minimal use of bipolar electrocautery to complete haemostasis was allowed. In our department, we use bipolar electrocautery at the lowest setting for coagulation for as short a time as possible, intermittently (at 30 W for a maximum of 2 s).

After surgery, children were monitored in our intensive care unit next to the operating theatre for 20 min, until they were awake and stable, then returned to the clinical ward. Children are encouraged to start early oral intake of clear liquids. The adenoidectomy patient is then supervised for approximately 6 h before discharge. Patients living within 1 hs distance from the hospital are discharged on the same day if no early postoperative bleeding occurs and if they have successfully restarted oral feeding.

Group A (continuous suctioning) enrolled 70 patients in whom continuous suctioning via a metallic probe with a perforated tip was used during adenoidectomy haemostasis. Group B enrolled 70 patients—in whom no continuous suctioning was used during haemostasis.

Suctioning method. We typically use a large metallic tube with a wide tip (approximately 1 cm in diameter) and multiple orifices at its end; the instrument is used both for suctioning from the nasopharynx and as a tongue depressor (Figure 1—the first 2 probes from left were used in our study). The suctioning pressure is set at a low level (150 mbar/hPa). Preferably, the suctioning from the child’s nasopharynx during adenoidectomy haemostasis involves keeping the probe’s tip at a few millimeters distance from the surrounding nasopharyngeal walls. This way, the metallic probe avoids direct contact with the mucosa and produces a local mucosal turbionary air flow through the nasopharynx, resulting in moderately lower local temperatures and thus stimulating vasoconstriction in the area. The duration of continuous suctioning is 3–5 min for each patient in group A.

Outcome variables. Two of the investigators measured and recorded the studied variables in all the patients enrolled in this study: total operative time (time was measured beginning with the first surgical gesture, after retraction of the soft palate, until adenoidectomy completely ended and all instruments were retracted from the operated child’s nasopharynx), haemostasis time (defined as the time elapsed after complete excision of the adenoid mass to completion of surgery, when surgical instruments are entirely withdrawn from the child’s mouth and nose), and the number of cotton packings used intraoperatively (in order to assess semiquantitatively the blood loss; a packing is considered used if at least 80% soaked in blood). The need for electrocautery use was defined as failure of haemostasis (over 10 min of haemostatic time and/or intense localised and persistent bleeding from the adenoidectomy area) and consequent use of bipolar electrocautery to complete haemostasis. The need for electrocautery use was noted for each group (yes/no answer). Early postadenoidectomy haemorrhage (occurring in the first 24 h postoperatively) in both groups was inferred from OR surgical protocols (yes/no answer) and was retrospectively included in our database. These variables, together with the age and sex of each patient, were used to complete our Excel database.

Choosing the studied sample size followed calculations performed by our statistical analyst and economic reasons. Moreover, it was the number of patients operated on by the 3 designated surgeons on 2 days per week during the study period of 6 months. A statistical analysis was performed on our database to compare the results between the two groups. The R project for statistical computing (R version 4.1.3), Welch’s Two Sample *t*-test, and Pearson’s Chi-squared test with Yates’ continuity correction were used to compare results between the two groups. A *p* ˂ 0.05 was considered statistically significant.

## 3. Results

A Welch Two-Sample *t*-test was used to compare variables between the two groups—group A (70 patients in whom we used continuous suctioning in order to control postadenoidectomy bleeding) (continuous suctioning group) and group B (70 patients with spontaneous postadenoidectomy haemostasis) (no suctioning group). In all the patients, adenoidectomy was performed with cold steel instruments and eventually bipolar electrocautery for bleeding control.

Analysing patients’ ages, distributions are similar between the two groups: the mean age in group A was 4.91 years (1–14 years) and in group B was 4.87 years (1–11 years). The difference was not statistically significant (*p* = 0.912), thus the two groups were similar and comparable. Group A consisted of 26 girls and 44 boys, while Group B consisted of 31 girls and 39 boys.

The mean operative time (duration of the adenoidectomy procedure) in group A (continuous suctioning group) was 11.48 min (688.57 s) (with variations between 5.57–22.49 min), while in group B it was 14.49 min (869.72 s) (with variations between 5.10 and 24.14 min). The difference was statistically significant (*p* = 0.00001733, t = −4.4988, df = 107.75).

The average haemostasis times were 8.67 min (520.54 s) in the A group (continuous suctioning group) (range between 3.44 and 13.00 min) and 10.97 min (658.06 s) respectively in group B (range between 2.17 and 23.29 min). The difference between the two groups was statistically significant (*p* = 0.0007169, t = −3.4899, df = 101.27).

A graphical representation of surgery duration with haemostasis, with continuous suctioning, and without suctioning is found in Figure 2.

The assessment of quantitative intraoperative blood loss by recording the number of cotton packings used showed an average of 6.04 for group A patients and 8.68, respectively, for group B patients. The difference was statistically significant (*p* = 0.00000007763, t = −5.8318, df = 93.687). A graphic representation of the number of packings used intraoperatively versus total operative time, with and without continuous suctioning, can be found in Figure 3.

Failure of haemostasis and the need for bipolar electrocautery application to the nasopharynx occurred in 4 patients in group A (4 out of 70 patients) and in 22 patients in group B (22 out of the total 70 patients). The difference was statistically significant (*p* = 0.0002202, X-squared = 13.65, df = 1) (Figure 4).

Early postadenoidectomy haemorrhage (occurring in the first 24 h after adenoidectomy) was noted in 3 cases in group A and in 2 cases in group B. This difference between groups was not statistically significant (X-squared = 0, df = 1, *p* = 1), regardless of the use of continuous suctioning (Figure 5).

The structure of the two groups of patients and the results of our study are summarised in Table 1.

## 4. Discussion

There are many studies in the literature regarding the haemostatic effect of different topical agents for bleeding control during adenoidectomy. Most authors agree that topical agents such as hydrogen peroxide, tranexamic acid, saline solutions, and vasoconstrictor/epinephrine dilutions applied locally to the nasopharynx seem to be the most effective in controlling haemostasis after such surgery [8,14,15,16,17,18,19]. Moreover, a multitude of active fibrin-based topical biomaterials and calcium alginate dressings are used on a large scale to control bleeding during adenoidectomy and accelerate mucosal healing [18]. All these topical agents involve supplementary costs, and their use is largely dependent on the ENT surgeon’s preferences and or experience and on the availability of materials from the hospital’s supply [9,18].

One study regarding the effect of saline solution irrigation during adenoidectomy haemostasis reported that saline solutions are effective haemostatic agents, especially at hot temperatures (50 °C). The haemostatic effect of hot saline solutions (at 50 °C) was significantly higher compared to the use of cold saline solutions (at 4 °C) [16,17].

We could not find any studies regarding the use of continuous suctioning—a simpler and costless method to stop bleeding during adenoidectomy that is readily available during surgery. No side effects were noted regarding cold air use. Studies investigating lowering the local temperature (for vasoconstriction) at the nasopharyngeal level during adenoidectomy haemostasis report using irrigation of the nasopharynx with cold (4 °C) saline solutions. Most authors agree that cold solutions are less effective in controlling blood loss after adenoidectomy compared with hot solutions [16,17].

Many studies have been conducted to investigate the effects of hypothermia on haemostasis in vitro and in vivo, and these have yielded contradictory results. One of these studies found that mild hypothermia during surgery significantly increases blood loss by 16% (4–26%) and increases the relative risk of transfusion [20]. Other studies conclude that mild hypothermia (temperature >33 °C) is safe from the standpoint of bleeding and might induce increased ADP-stimulated platelet aggregation and increased platelet reactivity (whereas deep hypothermia, i.e., temperature <33 °C, might decrease platelet aggregation). Under normal conditions, blood flow is maximal at the centre of the vessel, and platelets are marginalised to the periphery, close to the scene of injury, thus promoting platelet-endothelial interaction. Mild hypothermia increases the viscosity of blood, thus the marginalisation effect of platelets is more prominent in a hypothermic situation; Higher blood viscosity during hypothermia decreases blood flow velocity, which facilitates the formation of a platelet plug because the forces that tend to draw the platelet plug away from the vessel wall are decreased [18,19]. These pro-coagulative factors are not easily measurable in vitro but seem to be present during hypothermia in vivo [22,23].

As studies show, when the local body temperature drops below 37 °C, platelets become more predisposed to activation by thrombotic stimuli, an event known as priming. The enhanced ability of platelets to prime at peripheral blood sites, where temperatures are lower and the chances of trauma are higher, is thought to have evolved as a protective and adaptative effect against bleeding in humans, whereas more central body sites have greater protection against thrombosis [23].

On exposure to cold, some authors report a marked increase in the affinity of the postjunctional alpha-adrenoreceptors for norepinephrine, resulting in a powerful vasoconstriction and consequent cessation of blood flow to the perypheric tissues (this phenomenon, called hunting reaction was first seen in dogs, then in humans) [24]; the local effects of lowering the temperature to a traumatised region of the body (and consequent enhancing effects on haemostasis) differ from systemic effects of longer exposure of humans to low temperatures, as most of the authors report low temperatures to increase morbidity and mortality in cardiovascular and cerebrovascular patients (by increasing blood pressure via sympathetic nervous system activation) [25].

In the endeavour of offering some explanation for the haemostatis effect of continuous suctioning, we measured the local nasopharyngeal temperature in some of our patients. The protocol of our study did not include assessing the local temperature of every patient enrolled in this study. The mean local temperature we measured intraoperatively at the adenoid area after use of cold air suctioning for 3 min (the device set at 150 mbar) in some of our patients is 34.4 °C, corresponding to mild hypothermia; our study reports enhanced haemostasis in these children, corresponding to data from the literature regarding the effects of mild hypothermia on haemostasis.

Using a metallic suctioning tube at the patient’s nasopharyngeal level is not the only way to create locally mild hypothermia, as this effect (“cold air effect”) can be reproduced regardless of the suctioning probe’s material (plastic as well) and form.

After statistically analysing the data collected in our study and comparing the two groups, we found that the mean operative time during adenoidectomy, mean haemostasis time, and the mean intraoperative blood loss were significantly lower in children in whom continuous suctioning was used during haemostasis. Moreover, the need for electrocautery use because of haemostasis failure is higher in group B of patients in the absence of cold air suction use during adenoidectomy haemostasis. Absolutely no side-effects were noted in our patients in group A (continuous suctioning).

Choosing the studied sample size followed calculations performed by our statistical analyst. Moreover, it was the number of patients operated on by the three designated surgeons on 2 days per week during the study period of 6 months. The sample was small enough to be economically efficient and large enough to draw conclusions that warrant further extension of this study to larger samples of patients.

The mean operative time (duration of the adenoidectomy procedure) in group A (continuous suctioning group) was 11.48 min, while in group B it was 14.49 min; the difference is statistically significant (*p* < 0.05). According to our results, using cold air during adenoidectomy haemostasis reduces operative time by up to 3 min! On an individual level, shortening the total operative time by 3 min might seem unimportant in paediatric adenoidectomy; however, this means shortening the operation by 25%. These findings also prove that continuous suctioning can shorten the total operative time in adenoidectomy, which is the aim of the present study. Shortening the total operative time by 25% can gain importance in a department where the number of daily adenoidectomy procedures is high.

The average haemostasis times were 8.67 min in group A (cold air) and 10.97 min in group B. The comparison of these values is statistically significant (*p* < 0.05). According to our results, using cold air during adenoidectomy haemostasis reduces haemostasis time by up to 2 min!

The number of cotton packings used (reflecting in a semiquantitative manner the intraoperative blood loss) for group A patients was on average 6.04, while for group B patients it was on average 8.68. The comparison between these mean values is also statistically significant (*p* < 0.05). According to our results, using cold air during adenoidectomy haemostasis reduces intraoperative blood loss. Assessment of intraoperative blood loss using the number of packings used intraoperatively for each procedure is a semiquantitative, rapid, and easily reproducible method. Blood suctioned from the patient’s nasopharynx is collected in a large graded device; however, individual, exact measuring of blood loss (in mL) for every procedure is difficult in patients from group B, in whom we do not use suction at all.

Failure of haemostasis and the need for bipolar electrocautery application to the nasopharynx were recorded in four patients in group A (4 out of 70 patients) and in 22 patients in group B (22 out of the total 70 patients). The difference is also statistically significant (*p* < 0.05). According to our results, using cold air suction during adenoidectomy haemostasis reduces the need for bipolar electrocautery, avoiding its significant side-effects.

In terms of early postadenoidectomy haemorrhage (during the first 24 h after surgery), there was no significant difference between group A and group B. Thus, we can conclude that cold air use does not interfere with early postoperative bleeding after adenoidectomy.

The dimensions of the adenoid mass were not taken into consideration in this study as an independent factor, as studies show operative time or postoperative haemorrhage risk do not correlate with the grade of hypertrophy of the adenoid mass [26].

Limitations of this study could be the number of enrolled patients (the effects of cold air suction during adenoidectomy haemostasis could benefit from studies on a larger number of patients; however, the size of our sample group is large enough to warrant statistical analysis and offer a correlation hypothesis). The fact that this study is based on a single—centre experience—with a consequently uniform surgical technique and general anaesthesia. The method for assessing intraoperative blood loss through counting the number of cotton packings used for each surgical procedure is a semiquantitative procedure that could eventually be perfected.

## 5. Conclusions

Local continuous suctioning at the level of the nasopharynx during adenoidectomy haemostasis (via suctioning tube) can be a costless and useful tool to promote immediate haemostasis, as it significantly shortens surgery and haemostasis time, lowers intraoperative blood loss, and reduces the need for electrocautery use in cases of persistent intraoperative bleeding.

## Figures and Tables

**Figure 1 medicina-59-01534-f001:**
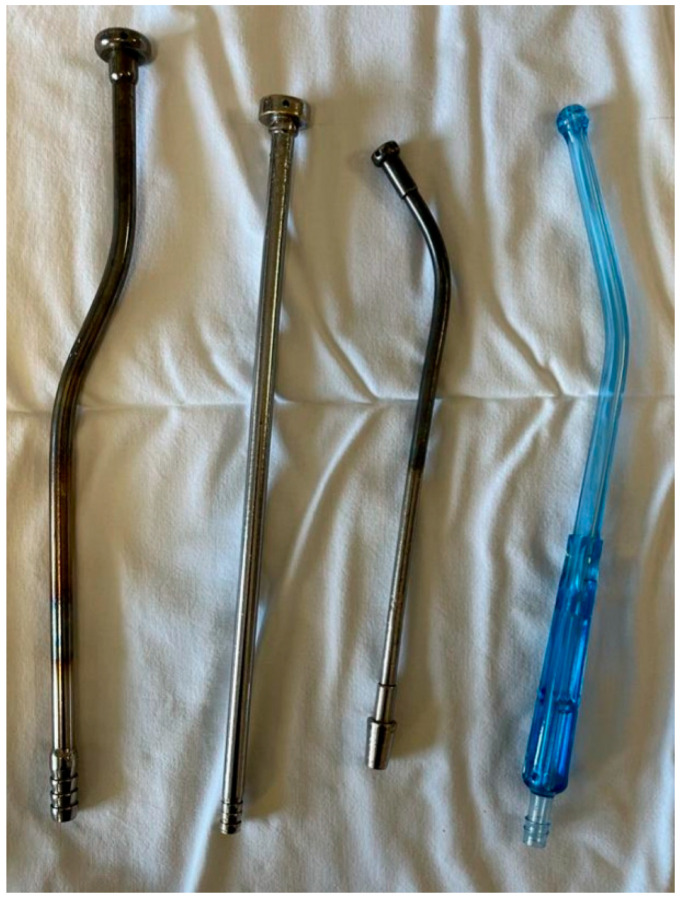
The metallic tubes used for continuous suctioning in our study. The pressure for suctioning is set at 150 mbar.

**Figure 2 medicina-59-01534-f002:**
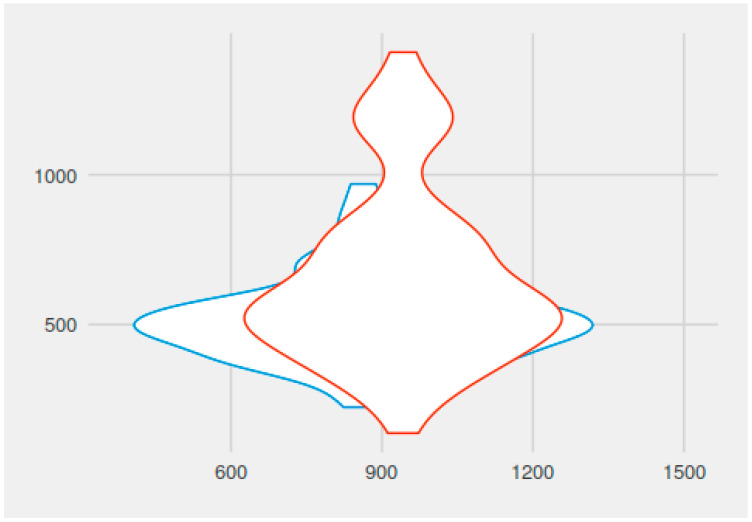
Duration of adenoidectomy procedure versus duration of haemostasis, with continuous suctioning (blue colour) and without suctioning (red colour).

**Figure 3 medicina-59-01534-f003:**
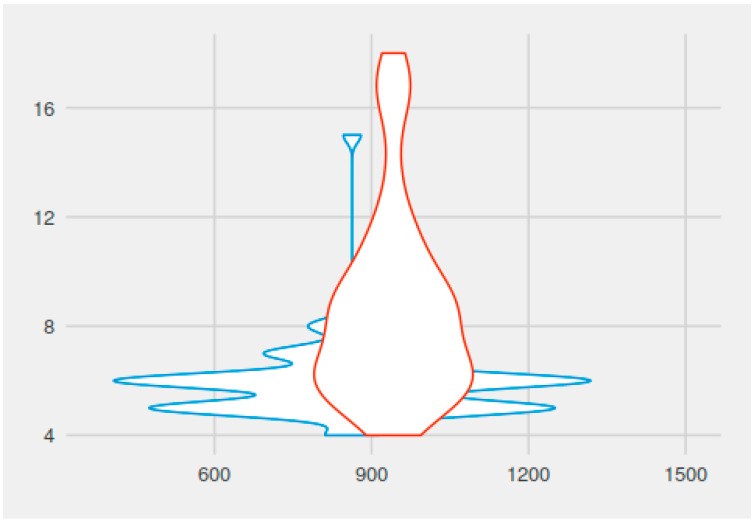
The number of gauze packings used intraoperatively versus total operative time, with (blue colour) and without suctioning (red colour).

**Figure 4 medicina-59-01534-f004:**
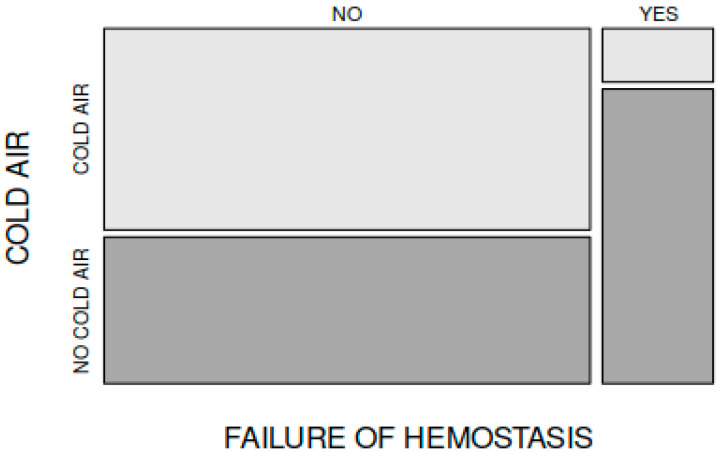
Failure of haemostasis and need for bipolar electrocautery use for haemostasis in groups A and B.

**Figure 5 medicina-59-01534-f005:**
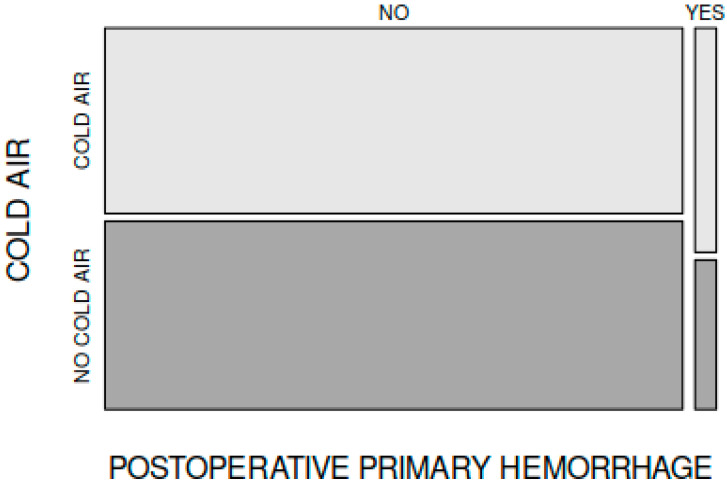
Incidence of early postadenoidectomy haemorrhage in groups A and B.

**Table 1 medicina-59-01534-t001:** Structure of the two groups of children, measured variables with comparison, and statistical significance of differences.

	Group A	Group B	Statistical Significance (*p* < 0.05)
Mean age of children	4.91 years (1–14)	4.87 years (1–11)	NO (*p* = 0.912)
Sex	26 F, 44 M	31 F, 39 M	
Mean operative time	11.48 min	14.49 min	YES (*p* = 0.00001733)
Mean haemostasis time	8.67 min	10.97 min	YES (*p* = 0.00007169)
Number of packings used	6.04	8.68	YES (*p* = 0.00000007763)
Failure of haemostasis/use of electrocautery intraoperatively	4/70	22/70	YES (*p* = 0.0002202)
Primary postoperative haemorrhage/use of electrocautery postoperatively	3/70	2/70	NO (*p* = 1)

## Data Availability

Data is available upon request to corresponding author.

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
