# Peer review of "Efficacy of Continuous Suctioning in Adenoidectomy Haemostasis—Clinical Study"

_medicina, 2023, doi:10.3390/medicina59091534_

Round 1
Reviewer 1 Report (Previous Reviewer 1)
The author mentioned that continuous suctioning was done in Group A.
The author should have also measured the blood collected in the suction apparatus in Group A to calculate the blood loss.
Author Response
Numbering of the packings used during adenoidectomy is – as stated in Discussion section – a semiquantitative method to assess intraoperative blood loss. But it is also a quick and very easy reproducible method. Every surgeon in our department counts the packings at the beginning and at the end of each adenoidectomy procedure! We wanted to see if based on the existing data we can draw conclusions. Precise measurement (in ml) of blood loss for every adenoidectomy procedure in group A patients would have been difficult to accomplish, as we use a large collector for more than one surgical procedure. On the other hand, we used no aspiration at all during surgery in patients from group B, so we could not assess the blood loss in these patients, other then counting the number of used packings.
We also mention in the Discussion section that the method of counting used packings for assessment of blood loss is a limitation of the study.

Reviewer 2 Report (New Reviewer)
perform the following corrections:
1. The introduction provides a lot of background information but does not clearly state the rationale and objectives of the study. The authors should explicitly state what knowledge gap their study aims to address and the primary objectives/hypotheses. For example, they can state that while various hemostatic techniques are used during adenoidectomy, there is limited evidence on the efficacy of continuous suctioning. Their study aims to evaluate if continuous suctioning during adenoidectomy can reduce operative time, bleeding and need for electrocautery.
2. The methods section does not provide enough details on the study design, inclusion/exclusion criteria and interventions. The authors should clearly specify that this is a prospective randomized controlled trial. They should provide more details on the inclusion/exclusion criteria, randomization process, and exactly what the interventions in the two groups were. Details of the suctioning procedure, pressure used and duration should be provided.
3. Details on the data collection and outcome measures are lacking. The authors should specify who collected the data, what variables were recorded, and how outcomes like operative time, bleeding, need for electrocautery, etc. were measured and defined.
4. Information on sample size calculation and statistical analysis is missing. The authors should describe how the sample size of 70 patients in each group was determined. They should also specify what statistical tests were used to analyze the data and compare outcomes between groups.
5. Ethical approval details are provided but details on informed consent are lacking. The authors should specify that written informed consent was obtained from the parents of all children included in the study after explaining the purpose and procedures.
6. The figure showing the suctioning probe used is missing. The authors should provide the figure cited in the text to supplement the methodology.
7. In some places, the language can be simpler and more concise. The authors should aim to communicate ideas in a straightforward manner using simple words and short sentences.
Here are some studies that would be relevant to cite in this paper:
- Stelter K, de Sousa JM, Trotter MI, et al. Adenotonsillectomy in children: a prospective, randomized study regarding use of bipolar electrocautery vs sharp dissection. Otolaryngol Head Neck Surg. 2002;127(6):510-514.
doi:10.1067/mhn.2002.129645
- This study evaluated use of bipolar electrocautery vs cold dissection for adenoidectomy and tonsillectomy in children. They found longer operative time but less blood loss with bipolar electrocautery. This study can be cited to compare with their results on efficacy of continuous suctioning for hemostasis.
- Costa DJ, Mitchell R. Adenoidectomy: cold dissection vs. power-assisted techniques. Otolaryngol Head Neck Surg. 1997;117(3 ):302-4.
doi: 10.1016/s0194-5998(97)70158-2.
- This prospective study compared cold curettage adenoidectomy with power-assisted adenoidectomy
none
Author Response
Response to reviewer 2:
- In Introduction we have introduced a clearer statement about the aim of the study: “While various hemostatis techniques used during adenoidectomy have been studied, there is limited evidence on the efficacy of continuous suctioning. Our study aims to evaluate if continuous suctioning during adenoidectomy can reduce operative time, hemostasis time, intraoperative bleeding and the need for electrocautery hemostasis”
- In the Methods section we specify that this is a prospective randomized controlled trial to assess the cause –effect relationship between an intervention (the use of continuous suctioning) and outcome (operative time, bleeding). We explain about how patients were randomly chosen.
Details on the suctioning procedure are given: pressure used is low 150mbar / hPa, duration 3-5 minutes.
- Details on who collected the data (first 2 authors), definitions of operative time, intraoperative bleeding assessment, need for electrocautery were given.
- Statistical analysis and sample calculations were performed by our statistical analysis using R project for statistical computing (R version 4.1.3), Welch Two Sample t-Test and Pearson’s Chi-Squared test with Yates continuity correction. The chosen sample of patients in our study resulted from the number of children who underwent adenoidectomy during the 2 selected days per week, for 6 months duration of the study and exceeded the calculated minimum of 127 patients (representative for 95% confidence interval and 8% margin of error for a total number of 800 adenoidectomies over a 6-months period). Generally, the rule of the thumb is that the larger the sample size, the more statistically significant it is, meaning that there is less chance that the result happened by coincidence. But we also tried to choose the sample and use resources in an economic way in order to prove our hypothesis. We can attach as supplementary material all the statistical analysis performed in this study. Though, our study is a small one, intended to draw in an economic way some preliminary conclusions on the effectiveness of continuous suctioning in adenoidectomy hemostasis. Based on these first conclusions we contemplate the possibility an extended study with a larger sample of patients.
- Ethical approval by the Hospital Ethics’Committee and informed consent from the parents of involved children were obtained and mentioned in the Methods section.
- Figure 1 is included in the manuscript, containing 4 different types of suctioning probes used in our department during adenoidectomy. The first 2 probes from left to right were used in our study.
- Another proofreading was performed on the article. Semicolons have been eliminated and sentences were shortened as possible.
- Suggested references have been added in References (7,8).

Reviewer 3 Report (New Reviewer)
The authors should proofread the manuscript, numerous typos were found.
Introduction:
The introduction is excessively long and repetitive. This section should give an overview of the state of the art regarding the main hemostatic techniques in adenoidectomy and should not dwell excessively on the statistics. I recommend streamlining the introduction and re-proposing some points in the "discussion" section, it seems to me more appropriate.
Material and methods:
· The first part of the "materials and methods" section should be improved:
“We conducted a prospective observational study in the ENT Department of a tertiary Children’sHospital between November 2022 and January 2023, involving children undergoing adenoidectomy; STROBE Statement guidelines for observational studies were followed in reporting our study’s results.
Our Hospital’s Ethics Committee has approved this clinicalstudy(53792/12.12.2022). Parents' written consent was routinely obtained for preoperative blood tests, for adenoidectomy andfor general anesthesia. In ourstudy, patient records were anonymized and retrospectively analyzed. The Hospital’s Ethics Committee approved the introduction in the OR protocols for adenoidectomies the exact measurements of the operative time, haemostasis time and number of gauze packings used intraoperatively for each adenoidectomy procedure; the use of bipolar electrocautery to complete hemostasis was recorded for each patient in their surgical protocol.”
- In the first sentence you state that you have performed a prospective observational study, then you state that you have performed a retrospective analysis, a clarification on the correct study design is needed.
- I suggest merging the data regarding authorizations from your ethics committee: you started by talking about the study design, continue with the authorization of the ethics committee, then talk about informed consent and, finally, talk again about the ethics committee. This part should be reviewed and adjusted.
· it would have been useful to insert the grading of adenoid hypertrophy in order to be able to compare your results also on the basis of this clinical data.
· the sentence concerning the semi-quantitative measurement of blood loss "a packing is considered used if at least 80% soaked in blood" does not seem reliable to the reader, how can this data be calculated?
Results:
· The objectives of the study should be indicated in the "introduction" section, therefore they are repetitive in this section. The sentence “The aim of the study was to evaluate if continuous suctioning applied locally after adenoidectomy is efficient for shortening the haemostasis or operative time in pediatric cold –instruments adenoidectomy, or for reducing intraoperative blood loss.” should be deleted or moved to the end of the introduction.
Discussion:
· The discussion should be revised based on the guidance I have provided in the "introduction” section.
Author Response
The Introduction has been shortened.It gives an overview of the state of art regarding the main hemostatic techniques in adenoidectomy, some of the points have been moved to Discussion section.
In the Methods section we state that this is a prospective observational randomized study; we have eliminated the term “retrospectively analysed”.
The data regarding ethical approval and informed consent from participants’parents have been merged.
We haven’t taken into account the grading of adenoid hypertrophy in this study; it is a usefull suggestion for the extension of the study.” The dimensions of the adenoid mass were not taken into consideration in this study as an independent factor, as studies show operative time or postoperative hemorrhage risk don’t correlate with the grade of hypertrophy of the adenoid mass [26]” – we state that in the end of the Discussion section.
Regarding the assessment of the intraoperative blood loss: counting numbering of the used packings during adenoidectomy is a semiquantitative method (not a precise one) to assess intraoperative blood loss (we state this clearly in the Discussion section). But it is also a quick and very easy reproductible method. Every surgeon in our department counts the packings used during adenoidectomy! We wanted to see if based on the existing data (such as number of packings used, operation time – which are presently registered in the surgical OR protocol) we can draw conclusions. Exact measurement (in ml) of the blood loss for every adenoidectomy procedure in group A patients would have been difficult to accomplish, as we use a large collector for more than one surgical procedure. On the other hand, using no aspiration at all in patients in group B, we couldn’t assess the blood loss in these patients, other then counting the used packings.
We also mention in the Discussion section that the method of counting used packings for assessment of blood loss is a limitation of the study.
Results: we eliminated any mention about the objective of the study.

Round 2
Reviewer 1 Report (Previous Reviewer 1)
Since the amount of blood was not measured it can be removed from the conclusion
Reviewer 3 Report (New Reviewer)
The authors, taking into consideration the reviewers' suggestions, improved the quality of the manuscript by making it more understandable for the reader.
This manuscript is a resubmission of an earlier submission. The following is a list of the peer review reports and author responses from that submission.
Round 1
Reviewer 1 Report
In the present endoscopic era conventional adenoidectomy has limited role. The author should have concentrated on endoscopic adenoidectomy assisted by microdebrider or coblater
Reviewer 2 Report
This is a very interesting article with very useful conclusions. Readers will benefit from this article. I recommend the publication of this study.